# Characterization of Sodium Alginate Hydrogels Reinforced with Nanoparticles of Hydroxyapatite for Biomedical Applications

**DOI:** 10.3390/polym13172927

**Published:** 2021-08-30

**Authors:** José Antonio Sánchez-Fernández, Gerardo Presbítero-Espinosa, Laura Peña-Parás, Edgar Iván Rodríguez Pizaña, Katya Patricia Villarreal Galván, Michal Vopálenský, Ivana Kumpová, Luis Ernesto Elizalde-Herrera

**Affiliations:** 1Procesos de Polimerización, Centro de Investigación en Química Aplicada, Blvd. Enrique Reyna No. 140, Saltillo 25294, Mexico; antonio.sanchez@ciqa.edu.mx; 2Centro de Ingeniería y Desarrollo Industrial (CIDESI), Sede Nuevo León 66629, Mexico; 3Engineering Department, Universidad de Monterrey, Av. Morones Prieto 4500 Pte., San Pedro Garza García 66238, Mexico; laura.pena@udem.edu; 4Biomedical Engineering Program, Universidad de Monterrey, Av. Morones Prieto 4500 Pte., San Pedro Garza García 66238, Mexico; edgar.rodriguez@udem.edu (E.I.R.P.); katya.villarreal@udem.edu (K.P.V.G.); 5Czech Academy of Sciences, Institute of Theoretical and Applied Mechanics, Prosecká 76, 190 00 Prague, Czech Republic; vopalensky@itam.cas.cz (M.V.); kumpova@itam.cas.cz (I.K.); 6Síntesis de Polímeros, Centro de Investigación en Química Aplicada, Blvd. Ing. Enrique Reyna No. 140, Saltillo 25294, Mexico; luis.elizalde@ciqa.edu.mx

**Keywords:** hydrogels, hydroxyapatite nanoparticles, mechanical properties, intermolecular interactions, micro-ct

## Abstract

In recent years, researchers working in biomedical science and technology have investigated alternatives for enhancing the mechanical properties of biomedical materials. In this work, sodium alginate (SA) hydrogel-reinforced nanoparticles (NPs) of hydroxyapatite (HA) were prepared to enhance the mechanical properties of this polymer. Compression tests showed an increase of 354.54% in ultimate compressive strength (UCS), and 154.36% in Young’s modulus with the addition of these NPs compared with pure SA. Thermogravimetric analysis (TGA) revealed that the amount of residual water is not negligible and covered a range from 20 to 35 wt%, and the decomposition degree of the alginate depends on the hydroxyapatite content, possibly due to the displacement of sodium ions by the hydroxyapatite and not by calcium chloride. Further, there is an important effect possibly due to the existence of an interaction of hydrogen bonds between the hydroxyl of the alginate and the oxygen atoms of the hydroxyapatite, so signals appear upfield in nuclear magnetic resonance (NMR) data. An increase in the accumulation of HA particles was observed with the use of X-ray microtomography, in which the quantified volume of particles per reconstructed volume corresponded accordingly to the increase in the mechanical properties of the hydrogel.

## 1. Introduction

Hydrogel polymers are attractive vehicles for the localized administration of pharmaceutical agents that can be released in a controlled manner. However, their low mechanical properties limit their use mainly to drug delivery and tissue engineering applications, and does not allow their application for orthopedic solutions [1]. Compared with metals, hydrogels do not show many secondary side effects, such as corrosion, stress shielding, wear particles that end up in the inflammatory response, pain, or dislocations, among others, complicating their application in spite of their great mechanical properties [2,3].

Sodium alginate (SA) is a common hydrogel polymer mainly extracted from brown seaweed or bacterial sources [4,5]. It is biodegradable and highly biocompatible and it is employed for tissue engineering purposes, as well as for protein and drug release [6,7,8,9]. Alginate is now known to be a whole family of linear copolymers containing blocks of (1,4)-linked β-d-mannuronate (M) and α-l-guluronate (G) residues. Alginic acid is a linear heteropolymer rich in l-guluronate, which forms strong but brittle gels, and d-mannuronate, which is weaker but more flexible [10]. Alginates that have low M:G ratios are expected to form strong but brittle gels and have been widely used for environmental, biomedical, pharmaceutical, food additive, and other industrial applications like bioremediation due to its high G-content [11].

The proportion of M and G residues and their macromolecular conformation determine the physical properties and the affinity of the alginate for divalent heavy metals by chelation [11,12]. With divalent cations, such as Ca^2+^, they form similar structures like the one presented in Figure 1b [13]. In this model, the M and G blocks form a three-dimensional structure where a divalent Ca atom is linked to two carboxylates of the alginate [14]. Further, this structure (dimer) allows for the entrapment of other divalent ions [15].

In order to improve the mechanical properties of SA, the addition of nanoparticles (NPs) has been studied. The reinforcement given by the NPs make them suitable materials for biomedical applications to support higher stresses [9]. Hydroxyapatite (HA) NPs, which are biocompatible ceramics with vast biomedical applications, have shown the ability to enhance mechanical properties. HA NPs are biocompatible ceramics that have a vast biomedical application; for example, HA increased the ultimate compression strength (UCS) of SA by 57.89%, according to the literature [8,9]. Additionally, HA can induce osteogenesis, showing great potential in tissue engineering [16,17,18].

In this research, SA + HA hydrogel composites with different concentrations of NPs were prepared and cross-linked with calcium chloride in order to create a compact hydrogel with a homogeneous consistency. The mechanical properties of the hydrogel were analyzed by performing compression tests. Finally, thermogravimetric analysis (TGA), nuclear magnetic resonance (NMR), and X-ray microtomography (Micro-CT) were also performed.

## 2. Materials and Methods

### 2.1. Materials and Synthesis of Hydrogels

Alginic acid sodium salt from brown algae (SA) (medium viscosity), HA with a molecular formula of Ca_5_(OH)(PO_4_)_3_, and calcium chloride (CaCl_2_) (pure reagent > 99%) were purchased from Sigma-Aldrich. For the elaboration of the hydrogels, different concentrations of HA NPs were used. The samples were named “SA(NPs)-(%NPs)” where (NPs) means the NPs used and (%NPs) represents the percentage of the NPs in the total weight between SA and NPs. The different samples prepared for this study are shown in Table 1.

The selected concentration of NPs was dispersed in distilled water (the amount was determined by the mass of the SA, 25 mL for every gram of SA) by magnetic stirring over 25 min. For better dispersion of the NPs, ultrasonication was applied for 5 min with a Cole-Parmer 500-Watt ultrasonic homogenizer with a frequency of 20 kHz. Then, the corresponding amount of SA was added to the blend and left in magnetic stirring overnight. Once the blend showed a homogeneous mixture, the hydrogel was cross-linked with 5 wt.% CaCl_2_ in a 0.5 M solution, where hydrogels went from liquid to solid state.

### 2.2. Hydrogel Composite Characterization: Thermogravimetric Analysis

Thermogravimetric Analysis (TGA) measurements were carried out in a TGA model Q500 (TA Instruments). The sample was kept at a constant temperature of 25 °C under a dry N_2_ atmosphere (50 mL/min), then the same specimen was heated from ambient to 600 °C at a rate of 10 °C/min and from 600 to 850 °C in an oxygen atmosphere at a rate of 50 °C/min. The TGA graph of each sample was obtained, indicating in them the percentage loss in weight against temperature, as well as its first derivative, which allowed the establishment of limits and maximum decomposition temperatures and peak temperatures (Tp).

### 2.3. Differential Scanning Calorimetry (DSC)

Differential scanning calorimetry was analyzed using a DSC-Q200, TA Instruments, New Castle, DE. The samples (∼12 mg) were transferred to an aluminum pan and heated from 10 °C to 200 °C with a heating rate of 10 °C min^−1^ under an oxygen-free nitrogen flow rate of 50 mL min^−1^.

### 2.4. Nuclear Magnetic Resonance

The sequence parameters of alginates were determined by ^1^H-nuclear magnetic resonance (^1^H-NMR) spectroscopy. Spectra were obtained from an NMR Bruker AVANCE III 500 MHz spectrometer operating at a field intensity of 11.7 tesla, using an HR-MAS probe and spectral windows of 7 kHz. The samples were analyzed in gel form and were diluted in deuterium oxide.

### 2.5. Mechanical Testing

Compression tests of hydrogels were performed in a multi-test dynamometer MTT 100, under 2 mm/min rate and increasing force from 0–100 lb, where the test was stopped after attaining the fracture of the hydrogel. The MTT 100 showed in real-time the displacement-force curve, which displayed a peak on the curve once the sample fractured, leading to the end of the test. The samples had average diameters of 25.75 mm. Strain-stress curves were plotted, determining the ultimate compressive strength of hydrogels with Equation (1), where F is the applied force and A the area of the composite samples. Strain was obtained using Equation (2), where ∆l is the difference between initial and final length of hydrogel, and l0 is the initial length or height of the sample. Accordingly, the secant line of the stress–strain behavior was used to obtain the Young’s modulus using Equation (3) for 50% strain, where σ_50%_ is half of the ultimate compressive strength (UCS) and ε is the compressive fracture strain.
σ = F/A(1)

ε = ∆l/l_0_(2)

E = σ_50%_ /ε
(3)

### 2.6. X-ray Microtomography (Micro-CT)

The differences in additions of HA NPs within the SA hydrogels were observed with the use of a patented computed tomography device “TORATOM” (Twinned Orthogonal Adjustable Tomograph) (EP 2835631). The established settings were: a voltage of 60.0 kV, a target power of 2.0 W, an acquisition time of 2800 ms, 3000 projections per rotation, and source to detector distance of 320.0 mm. A Dexela 1512 NDT detector (Varex Imaging Corporation, Salt Lake City, UT, USA) with a CsI scintillator and an active area of 1944 × 1536 pixels with a pixel pitch of 74.8 micrometers was employed.

The samples were positioned in a TORATOM as displayed in Figure 2 for the tomography procedure. To avoid drying risk, the SA hydrogel samples were introduced into a plastic sheath and kept moistened with distilled water during the tests.

According to the modality in which the structure of the specimens entered the field of vision, we settled on the following settings:−0% HA: source to object distance of 11.5 mm, to attain magnification of 27.8, with a pixel size of approximately 2.7 micrometers.−20%: source to object distance of 10.7 mm, to attain magnification of 30.0, with a pixel size of approximately 2.5 micrometers.−40%: source to object distance of 9.7 mm, to attain magnification of 33.1, with a pixel size of approximately 2.3 micrometers.−60%: source to object distance of 11.3 mm, to attain magnification of 28.4, with a pixel size of approximately 2.6 micrometers.

#### Isolation and Quantification of Particles of HA

Isolation and quantification of HA particles after corresponding micro-tomographies were performed with the use of the following software: a Fiji [19] and ImageJ2 [20] for filtration and thresholding; Pore3D [21] for filtering and segmentation of three-dimensional data; and VGSTUDIO MAX 3.2 for the selection and quantification of HA NPs. Particles were isolated from the rest of the material volume and their corresponding dimensional characteristics were measured. The following features were calculated: number of particles contained in the reconstructed volume; volume of particles (total sum of particle volumes (mm^3^)); total reconstructed volume of hydrogel (mm^3^); volume of particles per reconstructed volume: total sum of particle volumes related to the total reconstructed volume of the hydrogel; specific surface area (SV): sum of particle superficial areas regarding the total reconstructed hydrogel volume (mm^−1^). Micro-CT characterization allowed for the identification of particle size and distribution in the hydrogel composites. The particle size analysis was performed using the Histogram option on the MINITAB^®^ software.

## 3. Results and Discussion

Figure 1a shows SA linked by (1,4)- β-d-mannuronic (M) and α-l-guluronic (G), in which the schematic representation in Figure 1b suggests that there is a first interaction between hydroxyapatite and sodium alginate, modifying its morphology and causing lower crosslinking with Ca^2+^.

### 3.1. Thermogravimetric Analysis (TGA)

Table 2 displays the mass loss of the distinct integration contents of HA within the hydrogels at different temperatures.

Figure 3 displays the TGA analysis, in which we can observe four important transitions, the first from ambient temperature to 165 °C, due to the elimination of free water, while the second, between 165–230 °C, is due to the continuous loss of water before the decomposition. Hence, in addition to the free water, the samples also contain the other two kinds of water, like tightly bound water. The third transition, between 230–320 °C, is where the decomposition of alginate occurs; and the fourth transition, between 350–500 °C, is where the decomposition of the complex between alginate residues and its interactions with Ca^2+^ and the hydroxyapatite happen. On the other hand, in a comparative way, for the case of SAHA-0%, the transitions in the thermometric analysis of the alginate cross-linked with Ca^2+^ are < 135 °C, 3.74%; 135–200 °C, 4.48%; 220–320, 6.63%; 320–570 °C, 17.58%.

TGA analysis represents a technique that we can use to visualize the important hydrogel transitions, so the supramolecular interaction, a result of noncovalent interactions, including van der Waals interactions, electrostatic interactions, hydrophobic interactions, and chemical coordination between the host (alginate) and guest (hydroxyapatite) molecules, in addition to the crosslinking with Ca^2+^, can be determined. It also indicates the transitions of the amount of free and bound water that occur below a temperature of 230 °C. The molecular structure influences the absorption of water and can impact the swelling and the mechanical properties of the samples (Table 2).

### 3.2. Differential Scanning Calorimetry (DSC)

Figure 4 shows the results of the DSC analysis.

The properties of sodium alginate were affected by the intercalation with hydroxyapatite and subsequent linking with CaCl_2_. DSC thermogram displays endothermic peak behaviors, as shown in Figure 4. The endothermic peak around 115 °C might be caused by the loss of water from the matrix. Additionally, we found endothermic peaks around 180 °C that could show the dependence of fabrication of the hydrogels, the power of adsorption of water, and tightly bound water, that could be confirmed by the heating and cooling process in the analysis of DSC (Figure 4). In the same context, the molecular network of hydrogel contains molecular hydrogen bonds formed between carbonyl and hydroxyl groups that could be involved.

### 3.3. Nuclear Magnetic Resonance (NMR)

In Table 3, the ^1^H NMR signals ratio with the signal that appears in 1.25 ppm is shown; the specific signal at 1.25 ppm was taken as an arbitrary reference.

Following the results shown in Figure 5, the samples in this study were nearly or totally insoluble, which forced us to perform their characterization through solid-state NMR techniques. However, important spectral changes can be induced by the presence of an increased amount of hydroxyapatite, which can potentially affect the use of solid-state NMR in the structural investigation of insoluble alginates. The NMR data of the proton (^1^H) show that there is indeed an important interaction of sodium alginate and the hydroxyapatite that affects the crosslinking with divalent calcium ions. Additionally, upfield signals confirmed the finding.

Further, the amount of G block and M block in the ^1^HNMR spectra of alginate indicates clustering and partial separation of strongly hydrogen-bonded protons in the alginate gels (Figure 2). Hydrogen-bonding strength will increase as the disorder of the alginate chains increases [22].

Upon the formation of hydrogels, chemical shifts of protons in both host and guest molecules vary because their chemical environments change because of supramolecular interaction (noncovalent interactions) between the alginate and hydroxyapatite. Solid-state ^1^H NMR data show signals at δ 0.85–3.8 ppm precisely because of that interaction.

Figure 5 also shows the characteristic peaks in ^1^H NMR spectra obtained for different samples of alginate hydrogels, finding signals at δ 4.28 and 4.3 ppm that were attributed to the G residue hydrogen atoms linked to the C-2, C-3, and C-4 carbons. This signal disappeared in sample SAHA-60 due to a minor amount of alginate; peaks at δ 3.81, 3.80, and 3.7 ppm ascribed to the hydrogen atoms linked to the C-2, C-3, and C-4 carbons of the M residue. The alginate calcium/hydroxyapatite showed some additional characteristic peaks appearing at a range of δ 0.83–2.3 ppm, indicating that the alginate polymer microstructure was strongly dependent on the crosslinking calcium ion and the atoms of the hydroxyapatite structure [23,24,25].

### 3.4. Mechanical Behavior

Monotonic compression tests were performed until the fracture of the hydrogel was achieved, as shown in Figure 6. Additionally, Table 4 summarizes the results of the compression tests performed in all samples. The UCS hydrogels was increased by up to 354.54% with the addition of 60% HA. The Elastic Modulus also demonstrated important enhancements of up to 154.36% with 40%.

With regard to Figure 7, most of the samples have a similar non-linear behavior, thus E50% was calculated. As the NP concentration increased, the ultimate compressive strength also augmented. With HA NPs, the best UCS was found with the sample SAHA-60, with an increase of 354.54% compared to pure SA. This improvement is of high importance, as hydrogels commonly possess low mechanical properties that limit their application for load-bearing purposes.

The addition of HA NPs increased the maximum compression stress and Young’s modulus of alginate sodium hydrogels cross-linked with calcium chloride. Mechanical tests under monotonic compression have already shown an increase up to 2168% in ultimate compressive strength (UCS) and an increment of 770% in Young’s modulus, with the addition of these NPs compared with pure SA.

As we can see in the strain–stress curves shown in Figure 7, most of the samples had similar non-linear behavior, thus E50% was calculated where the curves had a linear performance once the second half was considered instead. As the NPs concentration increased, the ultimate compressive strength also augmented. We associate the enhancement found for the mechanical properties of hydrogels with the gradual inclusion of NPs, which increased pure SA hydrogel steadiness and served as reinforcement. Table 4 shows how UCS increased with the addition of NPs for each sample, as well as Young’s modulus behavior. HA NPs were able to increase the modulus by 154% with sample SAHA-40. In the case of SAHA-60, modulus decreased by 10.8% when compared to the case of SAHA-40; we consider this decline to be due to, as shown in Table 5, the number of particles diminished among these two HA conditions (the number of particles declined in 37.3%, under a 16% reduction in the reconstructed volume). While there was an increase in the UCS in SAHA-60, compared with SAHA-40, a considerable decrease in the number of particles caused a notable increase in strain conditions, producing a marked reduction in modulus. Nevertheless, regarding Table 5, in terms of the total volume of particles per reconstructed volume, the increase persisted among these two states.

UCS increase was of 355% in the case of SAHA-60, compared with pure SA-0 hydrogel with no NPs added, indicating a highest UCS of 0.95 MPa, with a Young’s modulus of 0.91 MPa. Our results for UCS surpass those obtained in other studies of SA hydrogels, with similar values for Young´s modulus [9]. Hence, the presented hydrogels have great potential to use as biomaterials for intervertebral discs and other biomedical applications where improved mechanical properties are required.

### 3.5. X-ray Microtomography (Micro-CT)

As presented in Figure 8, HA particles were observable within the transverse (top views), for the corresponding 0%, 20%, 40%, and 60% integration contents within the hydrogel.

A corresponding isolation procedure was attainable with the use of the software Fiji, ImageJ2, Pore3D, and VGSTUDIO MAX 3.2, in which, as indicated in Figure 9, we found (a) the reconstructed structure, (b) HA particles isolated from the whole reconstructed volume, and (c) the quantified terms of the volume of included particles.

Following the isolation procedure of HA particles for each SA sample, quantification was determined for each of them and, as observed in Figure 10, we recognized the accumulation increase of HA particles within each volume of the hydrogel in case.

Hence, Table 5 displays the corresponding quantification in terms of the number of particles, volume of particles, sample volumes, the volume of particles per reconstructed volume, as well as specific surface area (SV).

With the use of the corresponding tomography settings, as well as proper procedures for the isolation and quantification of HA particles, the accumulation increase of HA particles with the percentages initially planned for the strengthening of SA hydrogels was evidenced. Preliminary attempts at X-ray emission to the samples without a proper humidity conservation medium showed that the samples tended to dry and shrink, forcing the use of a plastic sheath and keeping them moistened with distilled water. Additionally, their geometries were properly reconstructable for the corresponding measurement of HA NP accumulation within their structure.

Following the mechanical properties results when increasing the content percentages of HA, we observed significant increases in terms of ultimate compressive strength (UCS), as well as Young´s modulus (E), for lapses among 0%, 20%, and 40%. Accordingly, for the same percentage lapses, we observed a considerable increase in HA particle accumulation following the use of micro-CT, as shown in Table 5, for results of the number of particles, volume of particles, volume of particles per reconstructed volume, and specific surface area. As presented in the results of UCS and E, for lapses among 40% and 60% HA contents within the SA hydrogel volumes, UCS values increased, although not under significant terms. Concerning E, its values even decreased from 1.02 to 0.91 MPa between these percentage lapses, meaning a 10.8% decrease. These differences accord from the contents evidenced with the use of micro-CT:

The Minitab software distribution of particle diameters analysis established that homogeneity of sizes was from 0 to 0.072 mm for 20 and 40% and 0 to 0.135 mm for 60%. Using the number of particles (N), mean (Mean), and standard deviation (StDev) of particle diameter, we acquired a measurement of what happens in the compound, as the particle diameter mean frequency establishes around 0.02 mm, regardless of the percentage of integration of particles in the hydrogel. This mean diameter also showed preservation in the nanostructured characteristic of the particles. Additionally, there was a bias to the right in the particle size distribution, presenting an accelerated increase in particle size and maintaining a gradual size reduction when surpassing the average of 0.02 mm.

The number of particles increased very considerably from 20% to 40% HA content, decreasing again drastically from 40% to 60%. Volume of particles, the volume of particles per reconstructed volume, and specific surface area all increased, although not significantly. The considerable decrease in the number of particles among 40% and 60% in HA contents, even when acquiring higher volumes and specific areas, may explain the tendency for attaining higher elastic properties in the 40% group in relation to the 60% group, causing a lowering of the value of E. This despite, as observed in the UCS and micro-CT results, the hydrogels effectively becoming tougher when increasing from 40% to 60% HA content.

Regarding tomographic analyses of HA particles in the development of hydrogels for biomedical purposes, to our knowledge, the only work reporting sodium alginate solutions with integration of hydroxyapatite particles was the one published by Oliveira et al. [26]. By employing a 7.25% (*w/v*) SA concentration as the vehicle and using a resolution of 10 μm, a homogeneous distribution of 35 wt % of microspheres was reported across the gel. In this regard, the author did not report particle dimensions’ measurements. Douglas et al. [27] introduced a bone cement by mixing reactive calcium-phosphate alpha tricalcium phosphate (α-TCP) particles with water to form hydroxyapatite (HA). With an adequate pixel size of 3.7 μm, the authors reported “the observation of a large number of small aggregates in the size range 103–104 μm^3^”. Another example was the acquirement of gellan-gum (GG) and hydroxyapatite (HA) hydrogels [28]; under a pixel size of 15.6 μm, the incorporation of different ratios of HA within the hydrogel, under 5%, 10%, 15% and 20%, was clearly shown with use of micro-CT. Henriques et al. [29], developed RGD-alginate hydrogel cross-linked with strontium Sr-doped hydroxyapatite microspheres. The author reported “an alginate-to-microspheres weight ratio of 0.35 and microspheres with an average diameter of 530 μm”.

New bone formation was evidenced under micro-CT, after 60 days of implantation. The use of a coating solution, with a composition of 52 wt% of water, 8 wt% of PVA, and 40 wt% of HA powder, was developed by Moreau et al. [30]. Although the author presented images of the coating under a pixel size of 35 μm, there was no quantification or particle size analysis.

Recent studies reported the effective use of SA/HA-based scaffolds as biomedical materials, in which they identified their possibility of use in bone defect repair [9], bone regeneration [31], and promotion of new cartilage generation [32]. As mentioned above, our study involved enhanced mechanical properties compared with others.

The improvement of the mechanical properties of SA hydrogel through nanotechnology opens the way to the investigation on orthopedic implants or lubricants in intervertebral discs, knees, hips, etc., depending on the most viable way of manufacturing and implanting the hydrogel. This study not only has a positive impact on the properties of the biomaterial but also on the field of public health that may lead to economic investments, without mentioning the benefits conferred upon the quality of life of patients suffering from orthopedic pathologies worldwide.

## 4. Conclusions

We adequately prepared SA in hydrogel samples reinforced with hydroxyapatite NPs (HA), which allowed us to improve the mechanical features. The elaboration of these hydrogels included different concentrations of HA NPs (NPs), represented in percentage volumetric contents of 0%, 20%, 40%, and 60%. HA NPs increased SA mechanical properties up to 0.95 MPa (355%) in ultimate compressive strength (UCS) for the case of SAHA-60, compared to the SA-0 hydrogel with no NPs added. Additionally, using Micro-CT, the proper isolation and quantification of HA particles was attainable, which showed an accumulation of HA particles in conformation with the gradual increase in the mechanical properties. The current hydrogels contain enhanced mechanical properties that surpass those obtained in other studies of SA hydrogels. Therefore, we affirm that these hydrogels will have a large capacity for biomedical materials usage, and near-future studies will confirm their applicability for intervertebral discs and other biomedical applications where improved mechanical properties will be required.

## Figures and Tables

**Figure 1 polymers-13-02927-f001:**
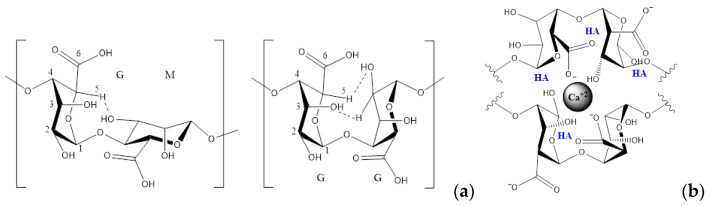
(**a**) Conformation possible of GM and GG diads due to the hydrogen-bonds. (**b**) Schematic representation of alginate-hydroxyapatite-Ca^2+^.

**Figure 2 polymers-13-02927-f002:**
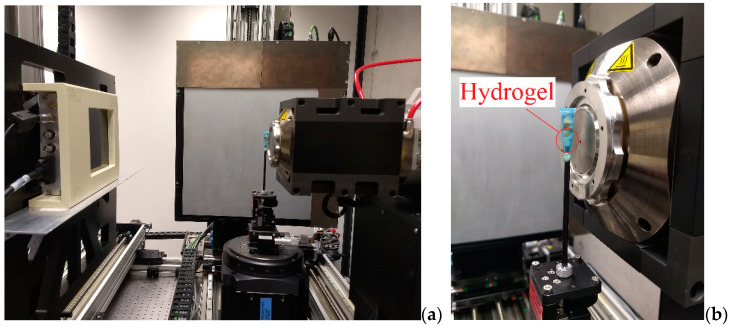
Positioning of the hydrogel for the tomography. (**a**) Overview of TORATOM and (**b**) positioning of the hydrogel.

**Figure 3 polymers-13-02927-f003:**
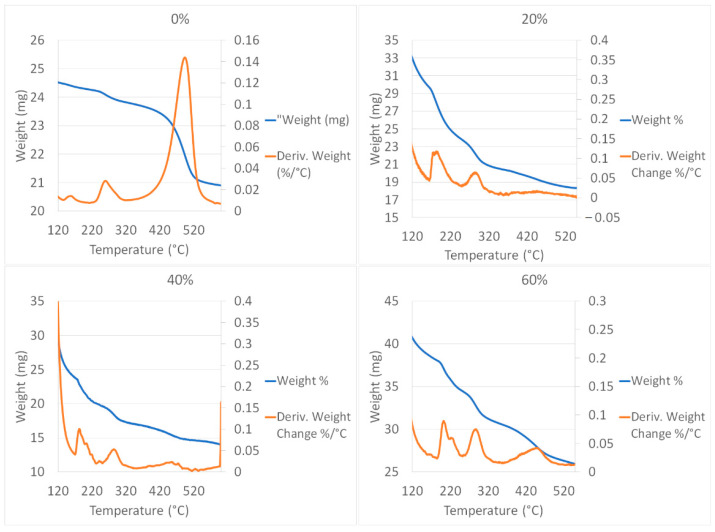
Thermogravimetric analysis (TGA) for 0, 20%, 40%, and 60% HA integration within the SA hydrogels.

**Figure 4 polymers-13-02927-f004:**
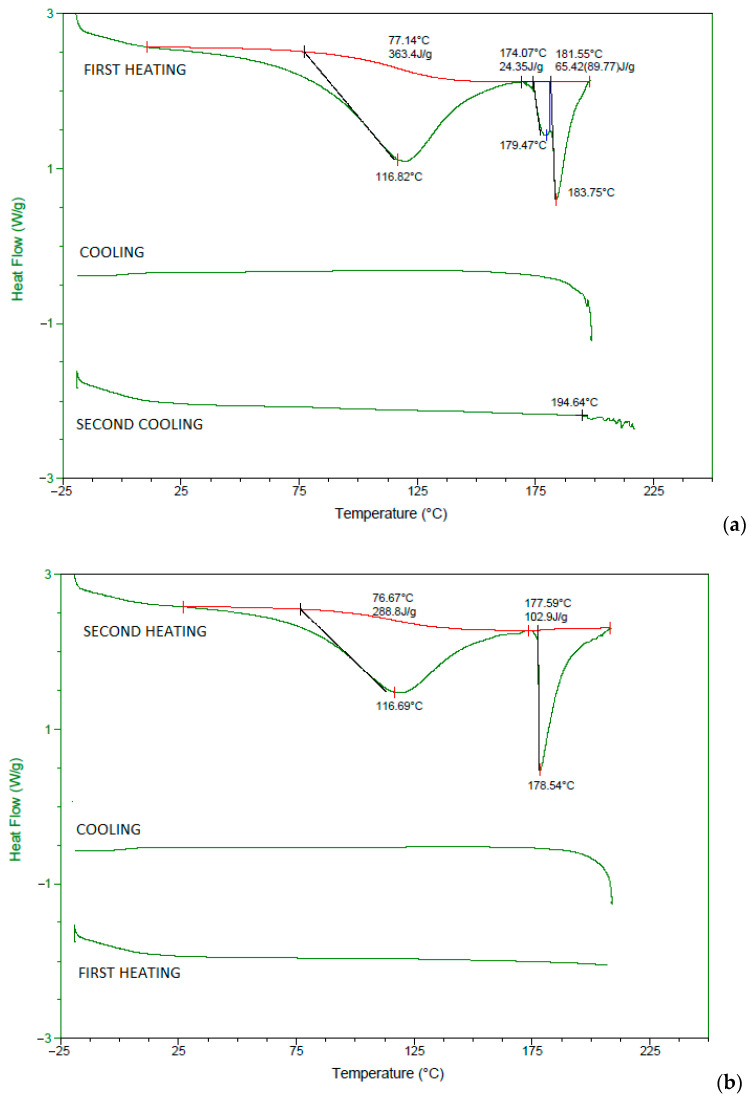
Differential scanning calorimetry (DSC) results for (**a**) 20%, (**b**) 40%, and (**c**) 60% HA integration within the SA hydrogels.

**Figure 5 polymers-13-02927-f005:**
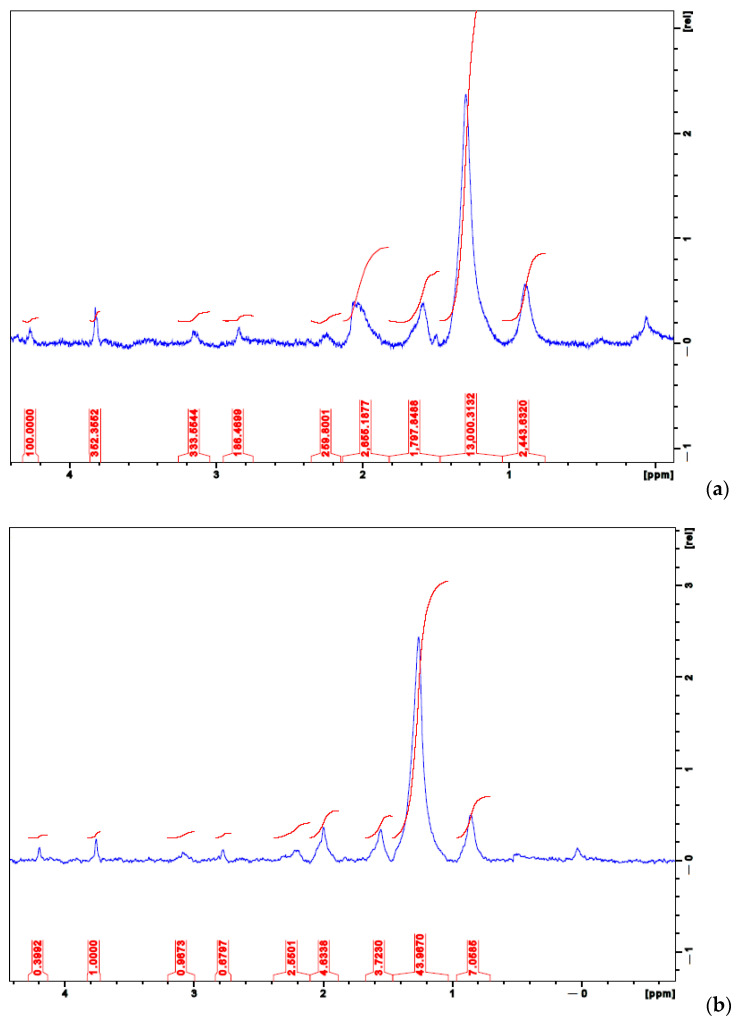
Nuclear magnetic resonance (NMR) results for (**a**) 20%, (**b**) 40%, and (**c**) 60% HA integration within the SA hydrogels.

**Figure 6 polymers-13-02927-f006:**
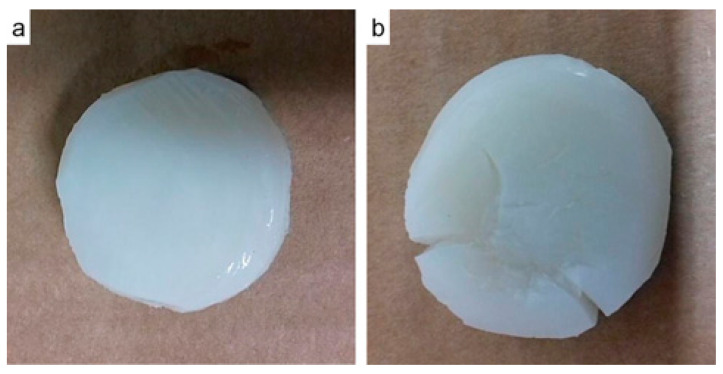
(**a**) Hydrogel before and (**b**) after fracturing by monotonic compression.

**Figure 7 polymers-13-02927-f007:**
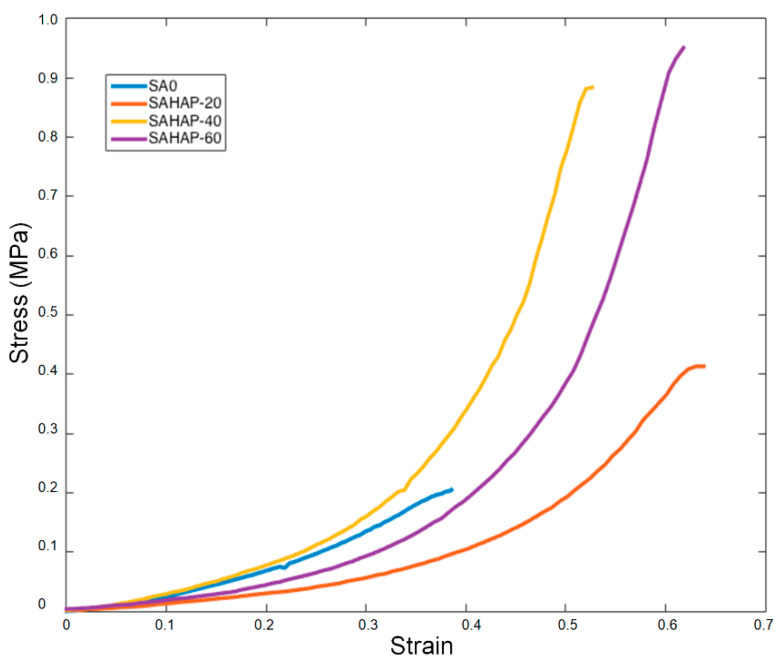
Stress–strain relationships for distinct HA integration within the SA hydrogels.

**Figure 8 polymers-13-02927-f008:**
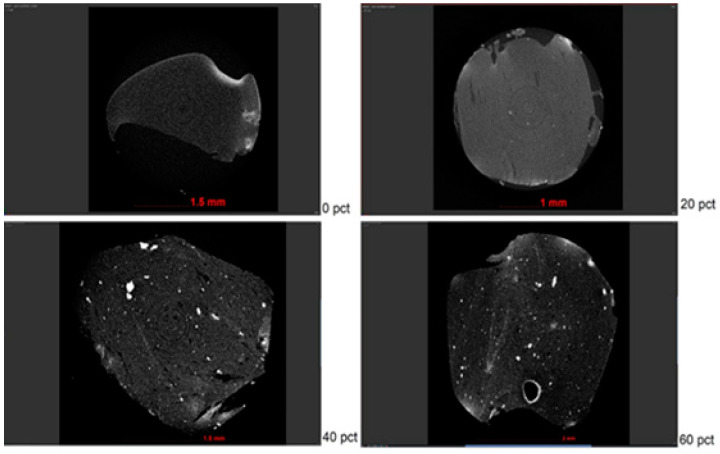
Transverse (top views) of SA hydrogels under differing integrations of HA (0%, 20%, 40%, and 60%).

**Figure 9 polymers-13-02927-f009:**
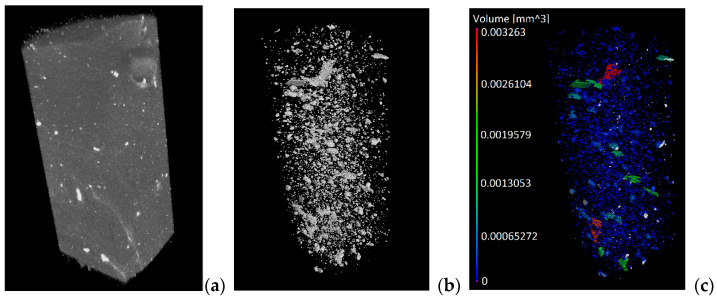
For a selected sample with 60% inclusion of HA particles, an overview of (**a**) the reconstructed volume, (**b**) isolation of HA particles, and (**c**) quantification of HA particles.

**Figure 10 polymers-13-02927-f010:**
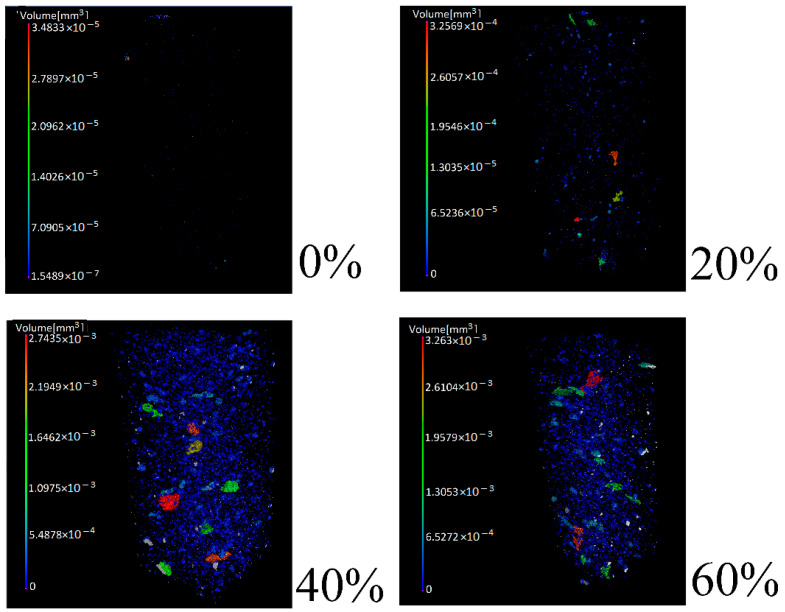
HA particles in SA hydrogels under differing integrations of HA (0%, 20%, 40%, and 60%).

**Table 1 polymers-13-02927-t001:** Sodium alginate hydrogels.

Sample	SA:NPs Ratio	CaCl_2_
SA-0	1:0	5 wt.%
SAHA-20	4:1	5 wt.%
SAHA-40	3:2	5 wt.%
SAHA-60	2:3	5 wt.%

**Table 2 polymers-13-02927-t002:** Mass loss at different temperatures.

Sample	SAHA-20 (%)	SAHA-40 (%)	SAHA-60 (%)
Mass loss < 165 °C	29.65	23.97	38.62
Mass loss 165–230 °C	5.23	3.90	3.90
Mass loss 250–320 °C	3.59	2.78	3.29
Mass loss 350–500 °C	2.29	2.57	4.66

**Table 3 polymers-13-02927-t003:** ^1^H NMR signal ratio with the signal that appears at 1.25 ppm.

Signals (ppm)	SAHA-20	SAHA-40	SAHA-60
0.85	0.19	0.16	0.16
1.25	1.00	1.00	1.00
1.65	0.14	0.8	0.12
2.00	0.20	0.10	0.14
3.80	0.03	0.02	0.04

**Table 4 polymers-13-02927-t004:** Ultimate compression test results and elastic modulus of hydrogel samples.

Sample	UCS (MPa)	Increase (%)	E (MPa)	Increase (%)
SA-0	0.209	-	0.401	-
SAHA-20	0.413	93.78	0.405	0.99
SAHA-40	0.883	322.49	1.02	154.36
SAHA-60	0.95	354.55	0.91	126.93

**Table 5 polymers-13-02927-t005:** Differing HA percentage contents in the SA hydrogels, in terms of the number of particles, the volume of particles, reconstructed volumes, the volume of particles per reconstructed volume, and specific surface area (SV).

	Number of Particles	Volume of Particles [mm^3^]	Reconstructed Volume [mm^3^]	Volume of Particles per Reconstructed Volume [mm^3^/mm^3^]	Specific Surface Area (Sv): [mm^−1^]
Hydrogel HA 0%	397	3.34 × 10^−4^	7.92	4.22 × 10^−5^	0.03
Hydrogel HA 20%	1396	4.55 × 10^−3^	7.71	5.91 × 10^−4^	0.17
Hydrogel HA 40%	12,956	6.94 × 10^−2^	6.87	1.01 × 10^−2^	1.87
Hydrogel HA 60%	8127	7.11 × 10^−2^	5.73	1.24 × 10^−2^	1.89

## Data Availability

Not applicable.

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
