# Peer review of "Characterization of Sodium Alginate Hydrogels Reinforced with Nanoparticles of Hydroxyapatite for Biomedical Applications"

_polymers, 2021, doi:10.3390/polym13172927_

Round 1

Reviewer 1 Report

In the manuscript, Sodium Alginate + Hydroxyapatite hydrogel composites, with different concentrations of NPs were prepared and cross-linked with calcium chloride, in order to create a compact hydrogel with a homogeneous consistency. The mechanical properties of the hydrogel were analyzed by performing compression tests. Additionally, thermogravimetric analysis, Nuclear Magnetic Resonance, and X-ray microtomography were also performed.

I have some comments for the manuscript.: 

L. 74 “In this research” – In the preset study

Inform the particle size and particle distribution in the hydrogel. Was the nanoparticle characteristic maintained? An analysis of a product sample by zetasinzer is essential to answer this question.

I suggest analyzing the sample by DSC

Figures 2, 3 and 4 are supplementary. I suggest they be deleted from the manuscript.

The first paragraph of topic 3.4 and figure 8 are related to the materials and methods topic. Reposition this information.

4.2. Include the figure corresponding to the NMR analysis.

Discussion

I suggest that the discussion not be divided into topics. If authors choose to leave it in topics, prefer to use a single section with results and discussion.

Conclusions

“Sodium alginate is a biodegradable and highly biocompatible hydrogel employed for 376 tissue engineering purposes.” This statement does not appear to be a conclusion of the study.

“we confirm that these hydrogels will have a large capacity for use as biomedical materials, especially for intervertebral discs and other biomedical”. How do authors ensure this capability? Additional tests on experimental models are needed for claims of this type. I suggest including as a perspective.

Author Response

Dear Professor.

Corresponding modifications and corrections have been added to the revised manuscript accordingly to your suggestions.

#Reviewer 1
In the manuscript, Sodium Alginate + Hydroxyapatite hydrogel composites, with different concentrations of NPs were prepared and cross-linked with calcium chloride, in order to create a compact hydrogel with a homogeneous consistency. The mechanical properties of the hydrogel were analyzed by performing compression tests. Additionally, thermogravimetric analysis, Nuclear Magnetic Resonance, and X-ray microtomography were also performed.

I have some comments for the manuscript.:

L. 74 “In this research” – In the preset study

Answer 1: Corrected in the manuscript.

Inform the particle size and particle distribution in the hydrogel. Was the nanoparticle characteristic maintained? An analysis of a product sample by zetasinzer is essential to answer this question.

Answer 2: We developed the study on particle size and particle distribution based on particle diameter information acquired from the X-ray computed tomography analysis.

I suggest analyzing the sample by DSC
Answer 3: Included in the manuscript.

Figures 2, 3 and 4 are supplementary. I suggest they be deleted from the manuscript.

Answer 4: Corrected in the manuscript.

The first paragraph of topic 3.4 and figure 8 are related to the materials and methods topic. Reposition this information.

Answer 5: Corrected in the manuscript.

4.2. Include the figure corresponding to the NMR analysis.

Answer 6: Included in the manuscript.

Discussion

I suggest that the discussion not be divided into topics. If authors choose to leave it in topics, prefer to use a single section with results and discussion.

Answer 7: Corrected in the manuscript.

Conclusions

“Sodium alginate is a biodegradable and highly biocompatible hydrogel employed for 376 tissue engineering purposes.” This statement does not appear to be a conclusion of the study.

Answer 8: Corrected in the manuscript.

“we confirm that these hydrogels will have a large capacity for use as biomedical materials, especially for intervertebral discs and other biomedical”. How do authors ensure this capability? Additional tests on experimental models are needed for claims of this type. I suggest including as a perspective.

Answer 9: Corrected in the manuscript.

Sincerely,

Gerardo Presbítero, PhD

Reviewer 2 Report

I have reviewed a manuscript entitled “Characterization of Sodium Alginate Hydrogels Reinforced with Nanoparticles of Hydroxyapatite for Biomedical Applications”. This work aims to evaluate the composite of sodium alginate (SA)/hydroxyapatite for biomedical applications. In terms of mechanical properties. I think it needs to address some comments before considering for publication:

Comment 1: please merge figures 1 and,5

Comment 2: please move 2-4 and 8 to the supporting information.

Comment 3: “Application” is misspelled in the title.

comment 4: please add the TGA graph to the manuscript.

Comment 5: please add a paragraph and explain the potential application of the SA/HA scaffold.

Comment 6: I would suggest adding the rheological properties of the composite before and after the addition of HA.

Author Response

Dear Professor.

Corresponding modifications and corrections have been added to the revised manuscript accordingly to your suggestions.

#Reviewer 2

I have reviewed a manuscript entitled “Characterization of Sodium Alginate Hydrogels Reinforced with Nanoparticles of Hydroxyapatite for Biomedical Applications”. This work aims to evaluate the composite of sodium alginate (SA)/hydroxyapatite for biomedical applications. In terms of mechanical properties. I think it needs to address some comments before considering for publication:

Comment 1: please merge figures 1 and,5

Answer 1: Corrected in the manuscript.

Comment 2: please move 2-4 and 8 to the supporting information.

Answer 2: Corrected in the manuscript.

Comment 3: “Application” is misspelled in the title.

Answer 3: Corrected in the manuscript.

comment 4: please add the TGA graph to the manuscript.

Answer 4: Included in the manuscript.

Comment 5: please add a paragraph and explain the potential application of the SA/HA scaffold.

Answer 5: Included in the manuscript.

Comment 6: I would suggest adding the rheological properties of the composite before and after the addition of HA.

Answer 6: Dear Professor. In this regard, we inform that since the hydrogel during the characterization stage was no longer in a liquid state (as shown in Figure 5), there was no possibility of developing rheological tests. For this reason, we performed mechanical testing of our composite hydrogels, which can provide valuable information on the properties and may help identify possible applications. However, we appreciate your suggestion and send you the rest of the corresponding corrections for your consideration.

Sincerely,

Gerardo Presbítero, PhD

Round 2

Reviewer 1 Report

The reviews performed by the authors significantly improved the quality of the manuscript, but I still have some considerations to make:

I am not sure if the results and discussion in this manuscript should be in separate topics. The manuscript has many figures and tables, and systematizing these data can help the reader. A solution to this would be to merge the two topics. I still find in the discussion that authors refer to figures and tables very categorically, which may also justify this junction. Thus:

1) My suggestion is that material and methods be joined.

2) Some results are not discussed satisfactorily (TGA, DSC, NMR). To review.

3) As these are thermogravimetric analyses, I suggest that the results/discussion of TGA and DSC come in sequence.

4) In the results, figure 3 looks out of place within the TGA topic (no comments are made for this figure). The same applies to the second paragraph of the discussion.

5) Figure 11 is supplementary and can be deleted. Interesting that the authors include in the text only the average particle size and distribution classification (unimodal, bimodal or multimodal).

Author Response

Dear Professor.

Corresponding modifications and corrections have been added to the revised manuscript accordingly to your second round suggestions.

I am not sure if the results and discussion in this manuscript should be in separate topics. The manuscript has many figures and tables, and systematizing these data can help the reader. A solution to this would be to merge the two topics. I still find in the discussion that authors refer to figures and tables very categorically, which may also justify this junction. Thus:

1) My suggestion is that material and methods be joined.

Answer 1: Corrected in the manuscript.

2) Some results are not discussed satisfactorily (TGA, DSC, NMR). To review.

Answer 2: Corrected in the manuscript.

3) As these are thermogravimetric analyses, I suggest that the results/discussion of TGA and DSC come in sequence.

Answer 3: Corrected in the manuscript.

4) In the results, figure 3 looks out of place within the TGA topic (no comments are made for this figure). The same applies to the second paragraph of the discussion.

Answer 4: Corrected in the manuscript.

5) Figure 11 is supplementary and can be deleted. Interesting that the authors include in the text only the average particle size and distribution classification (unimodal, bimodal or multimodal).

Answer 5: Dear Professor. Figure 11 was deleted from the manuscript, according to your kind suggestions.

Sincerely,

Gerardo Presbítero, PhD

Reviewer 2 Report

I think it is suitable for publication in the present form. 

Author Response

Dear Professor.

We sincerely appreciate your comments and support.

Best regards.

Gerardo Presbítero, PhD
